# Global Cultural Conflict and Digital Identity: Transforming Museums

**Tula Giannini** [1,*] **and Jonathan P. Bowen** [2,*]

1 Pratt Institute, New York, NY 11205, USA
2 Department of Informatics, London South Bank University, London SE1 0AA, UK
\* Correspondence: giannini@pratt.edu (T.G.); jonathan.bowen@lsbu.ac.uk (J.P.B.)

**Abstract:** This paper looks at key elements of global culture that are driving a new paradigm shift in museums causing them to question their raison d'être, their design and physical space, recognizing the need to accommodate visitor interaction and participation, and to reprioritize institutional outcomes and goals reexamining their priorities. As heritage sharing in online spaces reaches across national, political, and social boundaries on platforms and networks, this has been driven by museum engagement with Internet life during the pandemic. Museum relationships and interactions with communities both local and global continue to challenge core values and precepts, leading to radical changes in how museums define their roles and responsibilities. In this new cultural landscape, museums are responding to human digital identity in a tidal wave of human interactions on the Internet, from social media to online sharing of images and videos. This is revealing shared perspectives on cultural conflict as being tied to freedom of expression of one's heritage embedded in digital identity.

**Keywords:** art and technology; computational culture; cultural conflict; digital culture; digital identity; human behavior; interdisciplinary studies; museums

## 1. Introduction

For centuries, museums have been the collectors and curators of cultural heritage, displaying and protecting cherished objects in galleries of grand buildings. Like islands of national taste, museum objects are acquired through insider selection and communication with experts, curators, and professionals; meanwhile the public is allowed in to look, in ways that avoid interaction and participation, often charging fees to enter their rarified domain and having sparse communication with other institutions. Now, enter the digital world and the rapid expansion of the digitization of heritage objects and online access, shining new light on museum collections and art often hidden from view or seen on occasion in galleries with synthetic narratives by experts.

As heritage sharing in online public spaces reaches across national, political, and social boundaries on platforms and networks, driven by the museum world's obligatory engagement with Internet life during the pandemic, museum values and interactions with communities, from local to global are being tested to their core, causing radical changes in museums' societal roles and responsibilities. This new cultural landscape is causing museums to respond to human digital identity and the tidal wave of human interaction on the Internet, from social media to online sharing of images and videos, revealing global patterns of human activity and cultural expression.

Social justice movements from MeToo and Black Lives Matter to cultural conflict around heritage ownership, cultural identity, and colonialism have sparked claims for restitution and repatriation; making connections to climate change, the radical activism of the "Just Stop Oil" campaign is attacking art itself. What synthesizes these accelerating art and culture movements is their embrace of computational culture from digital identity, immersion in Internet communications, and spending the lion's share of work and play

time looking at screens, to using a myriad of devices such as smartphones, computers, watches, tablets, and televisions.

Yet digital art and the use of applications for navigation, information, interaction, and participation are still lacking in museums, acting as a type of digital divide between the museum and its audience/visitors. Artificial intelligence (AI) and machine learning (ML) have been growing in scope, becoming omnipresent in human activities, as humans regularly talk with chatbots for a growing number of tasks and, importantly, use their smartphone as their personal command center for a myriad of operations and activities. Thus, if upon entering the museum there are no digital dimensions to be found, many visitors are likely to feel isolated. As reality more than ever bows to digitality and virtuality [1]—intensified by the coming-soon Metaverse and the incarnation of virtual worlds, where human identity is being absorbed in avatar states of being—surely, it is time for computational culture to take its place in physical museums and connect with the global cultural community and ecosystem.

Judging from concrete actions of leading museums over recent years, we see that museum exhibitions and their narratives have changed dramatically, bending to social and political pressure and responding to the premise of the centrality of visitor experience, including the need for engagement and immersive exhibition experiences [2]. When the International Council on Museums (ICOM) states on its website, "Museums have no borders, they have a network", it acknowledges the global digital community. At its August 2022 meeting, the ICOM announced member approval of a new definition of museums. It brings into focus key elements that define a new path forward, underscoring the museum's relationship with the public, and sets the stage for user inclusiveness, participation, and interaction, advising museums to listen and respond to cultural conflicts with new awareness and vision [3]:

> "A museum is a not-for-profit, permanent institution in the service of society that researches, collects, conserves, interprets and exhibits tangible and intangible heritage. Open to the public, accessible and inclusive, museums foster diversity and sustainability. They operate and communicate ethically, professionally and with the participation of communities, offering varied experiences for education, enjoyment, reflection and knowledge sharing".

Two highly visible examples of cultural conflict are "looted" antiquities, as seen in the British Museum's long journey with the Elgin Marbles and the New York Metropolitan Museum's spurious acquisition of Cambodian sculptures. The latter were put on view using the Met's collection website, which brought these objects to the attention of the Cambodian government, causing them to make claims of repatriation and sparking global attention to Cambodian cultural identity, while inspiring a new awareness of Cambodian art, its value, and its extraordinary beauty, on view to the public via Internet networks and computational culture.

Bringing new perspectives to the digital culture landscape, we show that as the virtual world expands its reach into most aspects of human life, digital identity and human states of being meld into the new reality that we are now experiencing and dominates what seemed to be life divided between real and virtual. As AI and technology deepen our understanding of human memory and consciousness, we become mindful of how the mind captures and preserves cultural identity embedded in neural networks, extending to both real and artificial life. Herein lies the strength of our connections to cultural heritage, both tangible and intangible [4,5]. More and more, we observe globally how venues of digital communication and interaction bring higher levels of human awareness of the diversity of cultural heritage, as they intensify the central position of digital identity that ensures the flourishing of the global community.

## 2. Research Method

Museum studies is an established and developing field [6–8]. This paper adopts a new research methodology developed in the paper, which incorporates digital and physical re-

search, joining data from hands-on experience with digital capture and curation along with diverse digital resources, creating context and unique examples as we move from the particular to the general and past to present and contemplate the future. As such, it represents an emerging research methodology reflective of computational culture, in that it integrates digital research, evidence-based and primary source digital documentation, online humanities research, and contemporary and relevant experience, observation, and analysis.

A key aspect of this method is to engage the reader in the process of discovery and to elicit emotional and didactic responses to amplify awareness of diverse resources and the nature of cultural conflict and its complexity. We examine cross-disciplinary practice, where art and technology converge, and how this will continue to transform under the increasing pressure of computational culture [9,10], enabling hybrid lifestyles, positioned between reality and virtuality, while still respecting first-hand experience and perspectives. Importantly, the Internet offers a rich array of publicly available digital sources (blogs, online newspaper websites, etc.) that cross disciplines and institutional boundaries, creating new documentary juxtapositions able to construct fresh narratives, interpretations, and perspectives. Looking from the present to the future using a wide-angle interdisciplinary lens, we address emerging conflicts around centuries-old cultural differences and new questions of heritage ownership. Much of the material used for this research consists of digitized and born-digital documents and art in context captured by the authors, as opposed to existing journal publications.

Author observations of in-person and digital online experiences relating to digital technology, culture, and heritage are presented; observations were sourced in museum and gallery exhibits, visiting New York City, London, and Dubai, reading reviews, blogs, online newspapers, and seeing television shows, protests, and demonstrations during 2022. Google keyword searches enabled the researchers to identify opportunities that were trending online at the time, mainly in New York City and London, and to capture their observations, and to reflect on trends between digital technology, culture, and heritage drawing on their research over time. From onsite observations to diverse online resources, the researchers were able to inscribe a complex framework juxtaposing diverse understandings of cultural conflict in the broader landscape of heritage across the globe, transforming professional practice in an embrace of the complex rather than pre-packaged answers at a moment of rapid change. The rise of computational culture characterized by the pervasive use of artificial intelligence (AI) and machine learning (ML) is altering our understanding of data, its use and analysis. AI embeds itself across the Internet, from search engines and databases to social media and news. Research in this new hyper-data landscape poses limitations to its usefulness, bringing our attention to emerging methodologies that consider human perspectives, values, and ethics in the context of the emotional and sensual connections we experience through our cultural heritage and digital identity.

## 3. Digital Heritage and Identity

Standing at a crossroads between real and virtual 21st-century life, cultural conflicts are illuminated by online access to art collections, historical documents, and social narratives. We find ourselves exploring anew the past, rethinking the present, and revisioning the future of our global networked culture, using exciting new ways of knowing, doing, and experiencing [11]. Computational connections and technologies strengthen digital identity in the arts and the cultures they represent [12–14]. We are facing new decisions about how to make life's experiences fulfilling and enlightening, caught in a hybrid existence of physical and virtual activity, where, increasingly, the arts play a pivotal role that inspires and guides our choices. Connecting with our heritage is key to that journey of grasping our human identity. Yet, heritage and identity have become flashpoints of the arts in contemporary culture—the works, the objects, the narratives, and the history that they tell is expressed in the digital images we view online of diverse places and people, shifting from static to dynamic states, instantaneously a pure expression of cultural con-

flict spanning personal and national images of identity, seeking to reclaim art objects stolen or looted, making assertions of ownership.

Much has been written in the media about "culture wars", but what seems lacking in this discussion is the recognition that the conflicts we are experiencing are, in the first instance, being driven at an accelerated pace by advances in computing and technology; these are changing the nature and scope of communication, which now takes on global dimensions of human interaction and participation and includes both 2D and 3D, seeing, hearing, watching, learning, meeting, conferences, etc., so in essence, we are experiencing in real-time key facets of virtual life. This new virtual landscape of instant connections, comparisons, and data changes how we see the arts, ourselves and the world, advancing individual freedom and democracy, while the more socially advanced tech-savvy societies are the de-facto leaders, trendsetters, and iconic go-to societies where influencers of change show how to see the world in emerging contexts of identity and culture.

Whirlwind activity around a new social order is breaking up cultural and political silos, as humanity appears to be boarding a new freedom train where diversity, equity and inclusion are being embraced, while a relentless march toward life-changing technologies merges the natural and virtual worlds. This scenario, in the first instance, has put the U.S. at the center of this global revolution, noting that often, cultural conflicts are tied to the conflict caused by authoritarian power grabs that limit individual freedoms. There was a time when regressive policies and hostile actions could go unnoticed, but in the current world of Internet and network access, set in a computational framework, this becomes a diminished possibility. Our virtual vision sees beyond our physical place in real-time, and what was our smartphone now becomes our command center of experience and interaction across all aspects of life, art, and love. As we enter this new world amid revolutionary change, individuals feel more empowered to assert themselves in the global discussion.

### 3.1. Global Communication and Digital Identity

Few discussions of cultural conflict account for the impact of global communication via the web and Internet, from political platforms to social media and museum collection databases, etc. revealing hidden collections and objects not previously accessible or shared. Awareness—to know about, to hear about, to understand—is key and is a major contributor to the current evolution of cultural and human identity, as it converges with social justice movements and emerging technology, especially AI, ML, and natural language processing (NLP), which all help to facilitate access [12–14].

Further, the cultural conversation across the global ecosystem encourages new ideas, arguments, and, importantly, social engagement. Although culture wars have been around for decades, the convergence of these revolutionary factors in communication and access is unprecedented. We are living in a culture of the self, individual identity, and appreciation of difference and diversity, moving from the melting pot to the assertion and celebration of one's cultural identity. At the heart of this are the arts and the role of artists, who find themselves central to the search for self and to the feeling at one with the aesthetics coherent with their cultural identity—a powerful notion that influences many individual choices: fashion, definitions of beauty and sexual attraction, cuisine, visual preferences, music, and much more. It is in the foundational ties to one's cultural heritage that the essence of cultural aesthetics emerges, eliciting strong emotions and connections to cultural identity, see Figure 1.

### 3.2. Ukraine's Heritage in Time of War

As the war in Ukraine grinds on without resolution in sight, its horrific toll on human life and Ukrainian national heritage is seen most visibly in the destruction of its museums and the beauty of its natural landscape, both tied to the spirit and soul of being Ukrainian. At the heart of this cultural conflict is the human desire for freedom and a nation's thirst for self-determination. Ukraine has a long history of making important contributions to world culture, particularly in America.

*My Cultural Identity* by Tula Giannini

Reaching back centuries
to my cultural identity
I understand
who I am
from who I was

My look and sound
Not lost but found
The essence of being
The spirit of seeing
light at the end of my mind
to find my identity
touching reality
in the virtuality
of digital life

My past is my future
Its truth
in the computer
Heritage images
privileges who I am
what I see
My cultural identity
defines the me
in me

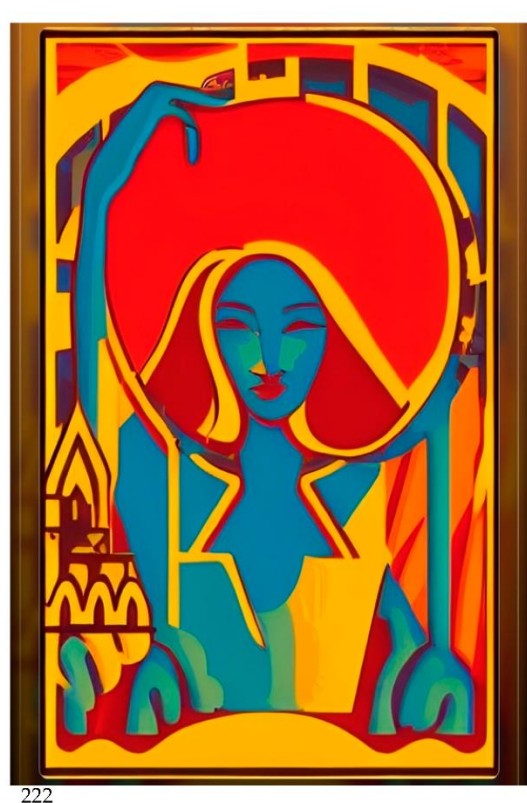

222

**Figure 1.** My Cultural Identity. Image by T. Giannini, generative art, October 2022.

Art becomes a target for destruction during war see Figure 2, owing to its power as a symbol of human expression and its ability to communicate the essence of human and cultural identity. In this respect, Ukrainian culture, although tied by history and geography to Russia, possesses a free and cherished spirit of its unique heritage—for example, the 20th-century composer George Gershwin, born in Brooklyn in 1898 (his father was born in Odessa), died in Los Angeles in 1937, and fellow-composer Leonard Bernstein, born in Lawrence, MA, 1918 (his Jewish Ukrainian parents were born in Rivne). Their careers show the powerful influence of heritage transmission and influence in the arts of America.

*The New York Times* writer Jason Fargo points to philosopher Anton Drobovych, who brings a fresh perspective on the war, and writes [15]:

> *Mr. Drobovych described Ukraine's culture as an explicit military asset. "It is contagious, and it spreads to the rest of the world", he told me at the barracks. "It seems that this truth and justice, which is violated, simply explodes in people who create art. And I think that's a huge part of why we're going to win this war".*

Although Ukraine's culture at the dawn of the 20th century seemed inextricably tied to Russia, seeing that many of the top Ukraine artists studied in Moscow, such as Gershwin and Bernstein, have become luminaries of the American cultural firmament, whose influence still holds sway, reminding us that cultural identity evolves through life's experience [16–18].

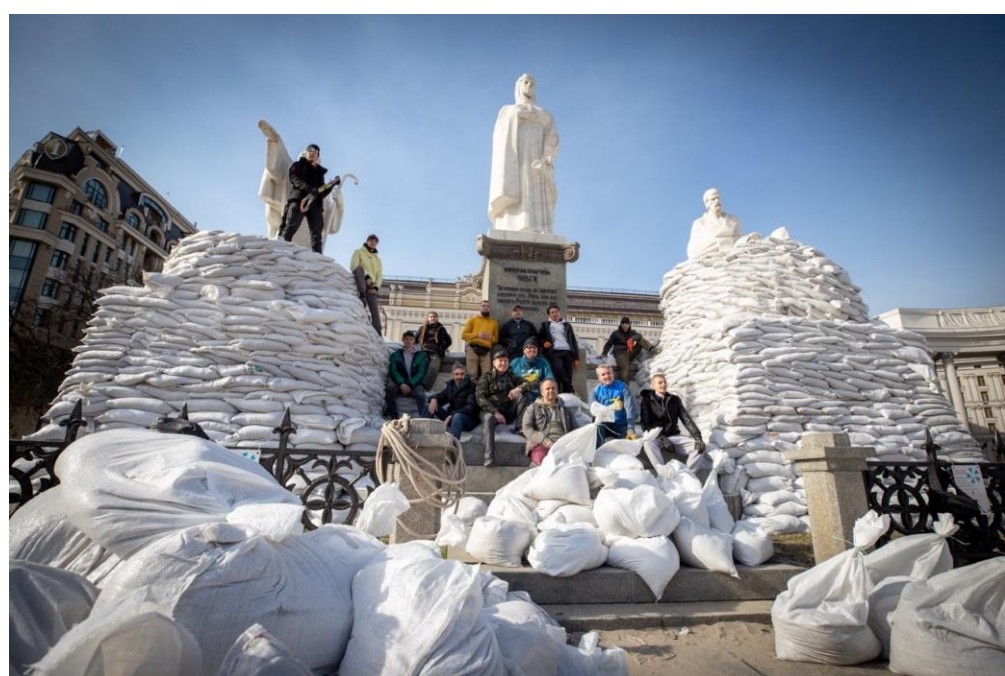

**Figure 2.** Princess Olha Monument during the Russian invasion of Ukraine; fortifying the monument, 18 April 2022. Wikimedia Commons. https://commons.wikimedia.org/wiki/File:Princess_Olha_Monument_during_Russian_invasion_of_Ukraine_(01).jpg (accessed on 30 January 2023).

*3.3. Digital Identity*

Digital identity, more than ever, is being embraced rather than hidden to blend into the mainstream to fulfill the notion of the "melting pot", as seen in fashion and design. From melting into a mass culture, we are blooming with imagination and diverse visions of self and a new sense of beauty and aesthetics. The melting pot has lost its allure; working together as individuals does not preclude differences of all kinds.

In [19], James Imam speaks to the divisive politics of nationalism in the arena of art and culture, recognizing that the arts sit at the center of digital identity and communication when, more than ever, the world shares a global computational culture that heightens awareness to the extreme. This is owed in part to the transition from a text to a visual culture in an emerging computational world, where reality and visuality are merged into a new global experience landscape, defining the way we live and work. The case of Italy's sharp turn towards the arts as a major part of their political core shows their deep sense of heritage, especially of Roman and Renaissance art, thus going back to the masters to guide the future and seemingly having a panic attack, fearing their loss of cultural identity and feeling the need to take measures to regain their place at the pinnacle of the arts' digital ecosphere [20]:

> *The dialogue between old and new was also a concern identified by Florence's social-democratic mayor, Dario Nardella. "Florence is not only an unbelievable capital of cultural heritage but it is also an important actor on the contemporary stage", Nardella told Artnet News. "This is our challenge: to be a big contemporary city, not only a city-museum, but also a living city capital of creativity, and innovation. This is the best way to keep the memory of the renaissance".*

*3.4. Cultural Heritage and Global Aggression*

Cultural heritage is always endangered by war. Witness the 2014 and then 2022 examples of the invasion of Ukraine, initially Crimea, by Russia. This situation was reflected in Expo 2020, held in Dubai during 2021–22, delayed due to the Covid pandemic. The Expo symbolizes a microcosm of the real world, with nearly every country in the world

represented through its own pavilion. During Expo 2020, Vladimir Putin decided to invade Ukraine on 24 February 2022. Suddenly the world was set in turmoil. The effect on the Ukrainian and Russian pavilions at Expo 2020 is interesting to note. At the Ukrainian pavilion, a man stood holding a large Ukrainian flag (see Figure 3). Inside the pavilion, near the entrance, a life-sized display stand, showing Ukraine's president, Volodymyr Zelenskyy, with the title, "Stand with Ukraine", and suitable for selfies, was erected. All the computer display screens were changed to show the blue and yellow Ukrainian flag with the hashtag #StandWithUkraine (see Figure 4). Small Ukrainian flags and carefully tied wrist ribbons in blue and yellow were issued to visitors. At peak times, there were queues outside the pavilion.

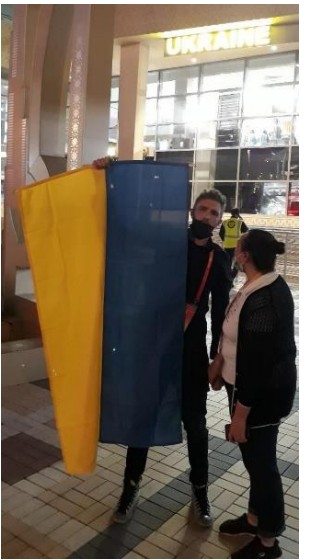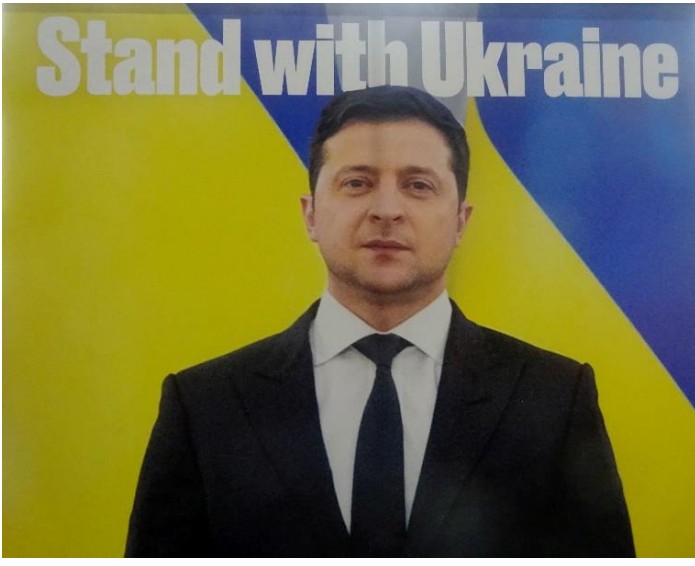

**Figure 3.** The Ukrainian pavilion at Expo 2020. (**left**): Displaying the Ukrainian flag outside the pavilion. (**right**): Poster stand of the Ukrainian president at the entrance.

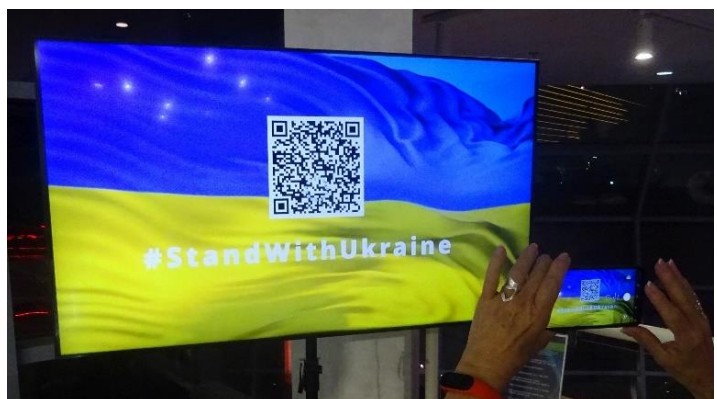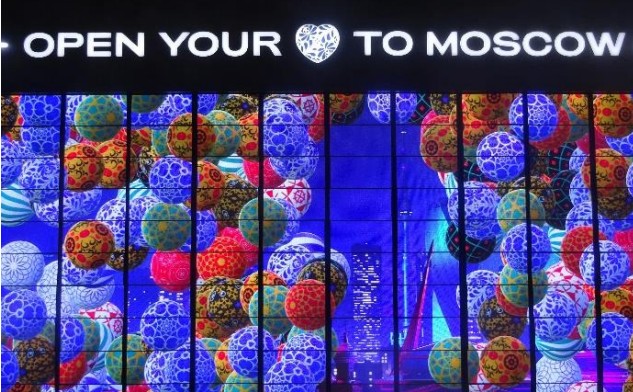

**Figure 4.** The contrasting Ukrainian and Russian pavilion displays at Expo 2020. (**left**): Display screen with #StandWithUkraine hashtag. (**right**): Ironic display in the Russian pavilion.

Meanwhile, at the Russian pavilion, it was as if nothing had happened. The explicitly stated theme was "cooperation", a chilling thought in light of the "Special Military Operation" set in progress by Putin. Displays inside the pavilion included headlines such as "From Russian with Love" and "Open your ♥ to Moscow" (see Figure 4), highly ironic in light of the actions taking place concurrently in Ukraine. It was as if everything was perfectly normal in the real world. Outside the Russian pavilion, security personnel discretely ensured that no demonstration of feeling could be shown, a reminder that the exhibition

itself was not in a country where dissent is tolerated, even if in a less extreme manner than in Russia itself.

Putin's attitude to cultural heritage became increasingly obvious as the war in Ukraine progressed. Ukraine and Russia have both signed the *Hague Convention for the Protection of Cultural Property in the Event of Armed Conflict* (known as the 1954 Hague Convention), which was created to help safeguard cultural heritage during armed conflict. This states that [21]:

> *"... any damage to cultural property, irrespective of the people it belongs to, is a damage to the cultural heritage of all humanity, because every people contributes to the world's culture..."*

UNESCO is primarily responsible for the monitoring of compliance and has maintained a list of cultural sites verified as damaged and/or destroyed during the 2022 Russian invasion of Ukraine (started 24 February 2022). As of 23 December 2022, UNESCO has verified damage to 231 sites since the start of the invasion. These include 11 libraries, 19 monuments, 18 museums, 81 buildings of historical/artistic interest, and 102 religious sites [22].

Memorial museums are important for developing empathy in visitors [23]. An example of a completely destroyed memorial museum is the National Memorial Museum of Hryhorii Skovoroda in the Kharkiv Oblast. This celebrated and memorialized the Ukrainian philosopher Hryhorii Skovoroda (1722–1794, aka Gregory Skovoroda), also a composer of liturgical music, a poet, and a teacher, who followed the philosophical Socratic tradition at the time of the Russian Empire. On 6–7 May 2022, the building was destroyed by Russian shelling and a subsequent fire (see Figure 5) [24]. Clearly, the 1954 Hague Convention is not a priority for Putin.

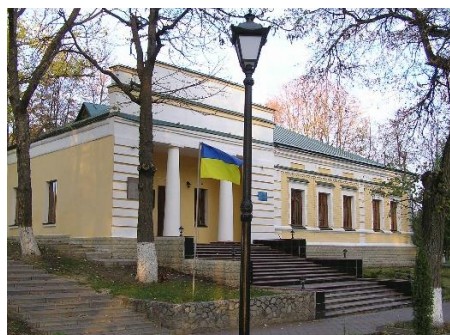 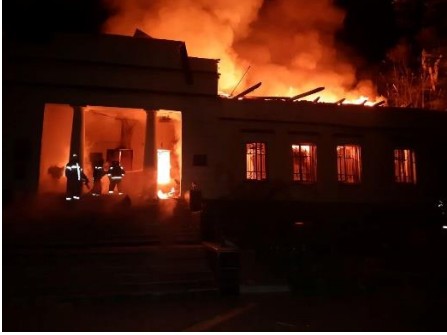 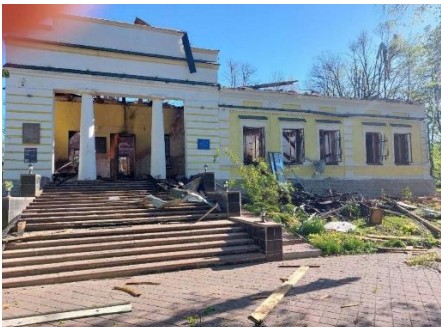

**Figure 5.** The National Memorial Museum of Hryhorii Skovoroda in Kharkiv Oblast, Ukraine. (**left**): In 2011. (**centre**): On 6 May 2022. (**right**): On 7 May 2022. Wikimedia Commons. https://commons.wikimedia.org/wiki/Category:National_Memorial_Museum_of_Hryhoriy_Skovoroda_(destroyed) (accessed on 30 January 2023).

Recording the existence of cultural heritage digitally is one way to protect it from complete destruction by those with no interest in it. Cultural heritage is closely connected with cultural identity [25], and digital preservation can help with restoration and reconstruction after war [26]. There are efforts to use 3D technology to preserve Ukrainian cultural artifacts in a digital archive, stored away from Russian attacks [27].

## 4. Heritage Repatriation, Restitution and Climate Activism

As old hierarchies disappear, we find museums engaged in a battle being hard-fought around heritage ownership and human cultural identity on a global scale, and no country or culture can hide. A newly emerging cultural order demands respect and recognition in the West of non-Western cultures by way of restitution, to rebuild their heritage, present to past, as we form cultural frameworks for a computational world.

*Heritage Activism: From "Stop the Oil" to Nazi Loot*

"Just Stop Oil" represents a growing number of disruptive actions in museums in the name of climate change across Europe and the UK. Funded by The Climate Emergency Fund in California and the Italian-based Ultima Generazione, the protesters claim they want to draw attention to the urgency of the climate change movement, although their aggressive action attacking art is at odds with museums' heritage preservation and conservation goals [28].

Van Gogh's *Sunflowers* (see Figure 6) was purchased in 1987 at Christie's for $40 million, a sale being sued in Federal court in Chicago by the heirs of Mendelssohn-Bartholdy [29]; it is doubtful that the protesters were aware of the painting's ties to Nazi loot, and museums are experiencing a wave of "Stop the Oil" protestors. The painting was owned by Paul von Mendelssohn-Bartholdy, a prominent German-Jewish banker and art collector (born 1875, died 1935 in Berlin) who had been "coerced" by the Nazis to liquidate his art collection in 1934, dispersing his collection of masterworks to museums and collectors. Tracing his family heritage shows that he was related to the great German-Jewish composer Felix Mendelssohn (1809–1847). His sister Fanny (1805–1847) was one of the few revered women composers of the period, descendants of Moses Mendelssohn (1729–1786), a leading German-Jewish philosopher of the Enlightenment focusing on metaphysics, aesthetics, and religion see Figure 7.

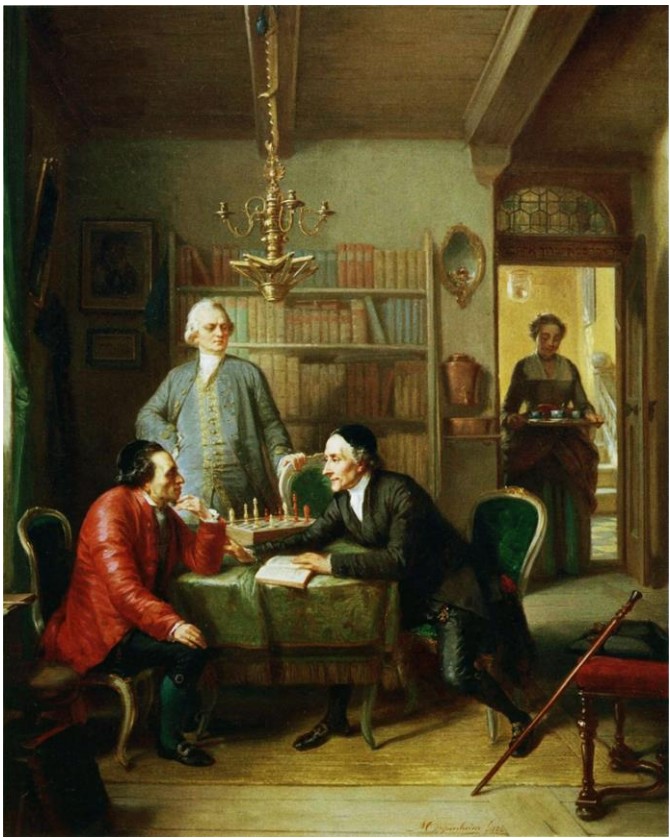

**Figure 6.** Gotthold Ephraim Lessing, writer and dramatist, and Johann Kaspar Lavater, Swiss theologian, as guests in the home of Moses Mendelssohn, 1856 painting by Moritz Daniel Oppenheim (1800–1882).

In the matter of restitution, the National Gallery of Art, Washington. D.C., and the Kimbell Art Museum announced that they had repurchased the Joseph Mallord William Turner painting *Glaucus and Scylla* (1841) see Figure 8, which the museum had previously returned to the heirs of John and Anna Jaffé after an investigation concluded that the painting had been unlawfully seized by the pro-Nazi Vichy regime in France in 1943 [30]. The painting was purchased at Christie's, New York, for a hammer price of $5.7 million.

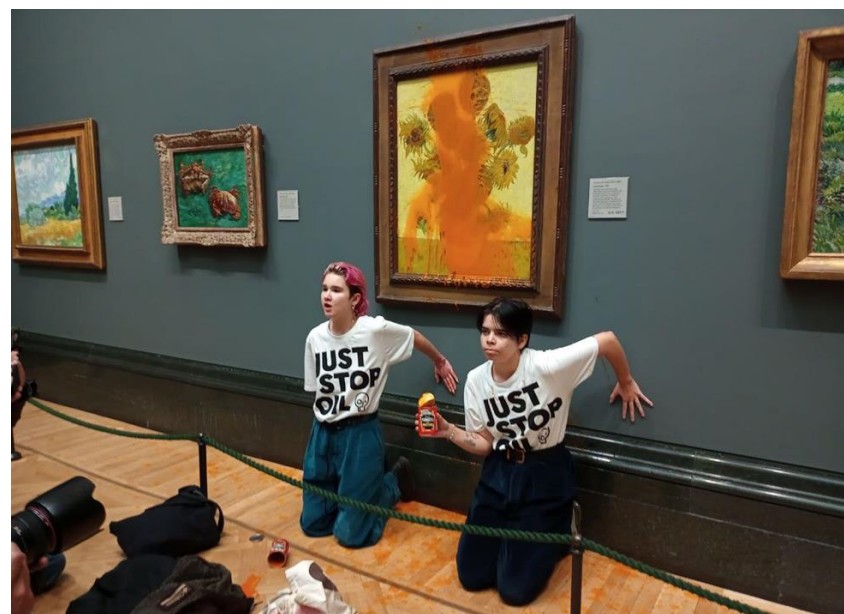

**Figure 7.** Just Stop Oil—Attack on Van Gogh' painting Sunflowers in the name of climate activism. "Two protesters who have thrown tinned soup at Vincent Van Gogh's famous 1888 work Sunflowers at the National Gallery in London, Friday, 14 October 2022. The group Just Stop Oil, which wants the British government to halt new oil and gas projects, said activists dumped two cans of Heinz tomato soup over the oil painting on Friday, 14 October 2022". (Just Stop Oil via Associated Press). Wikimedia Commons. https://commons.wikimedia.org/wiki/File:Just_Stop_Oil_National_Gallery_14102022.png (accessed on 30 January 2023).

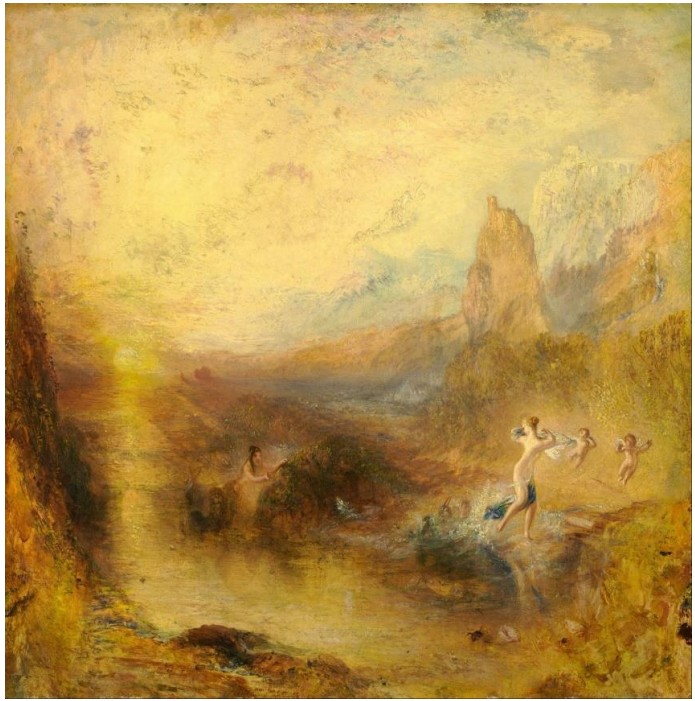

**Figure 8.** Joseph Mallord William Turner, *Glaucus and Scylla*, 1841. Seized in 1943 by the Nazis from the art collector Paul von Mendelssohn-Bartholdy, later acquired by the Kimbell Museum, then restituted to the Mendelssohn-Bartholdy family heirs, auctioned at Christie's, and repurchased by the Kimbell Museum for US$6,424,000 after restitution to heirs of John and Anna Jaffé, 18 April 2007 [31] https://commons.wikimedia.org/wiki/File:Joseph_Mallord_William_Turner,_%E2%80%98Glaucus_and_Scylla%E2%80%99,_1841.jpg (accessed on 30 January 2023).

## 5. Digital Identity and Heritage

At the heart of heritage and identity is claiming the past to create the present, while raising self-awareness and awakening a new sense of the self in our global digital landscape as it morphs from individual and institutional siloes to complex cultural identity frameworks. In the spirit of shared experience, we navigate a path of participation in global cultural networks, sparking new sociocultural movements where arts and heritage meet. The underlying and critical change is that awareness is inextricably linked to access and knowledge, why things exist and knowing where and why. This speaks to computational culture facilitating our being able to see inside museum and gallery collections. As we move further into the world of digital 3D imagery, the artifact comes to life, and with VR and AR, visitors can feel an immersive presence and experience of the object. These experiences, leading to discoveries of objects hidden in storage, raise critical questions about ownership tied to cultural identity and the power of human connections with heritage artifacts [32]. Being removed from their place of origin, the culture in which they were born, the narrative and interpretation of these artifacts, separated from their lived experience of their cultural heritage, leaves open the need to be redressed and for narratives to be rewritten. From the 2014 Black Lives Matter (BLM) protests at the Mall of America, a new movement emerged around cultural identity, leading to the repatriation of museum objects with the goal of returning them to their original heritage contexts.

From artifacts residing in the museum to the global Internet stage, their provenance is increasingly in the spotlight, raising questions of ownership. Large Western museums house massive collections of which some objects are on exhibition while others reside in storage, many never having been on view. The digitization of museum collections, particularly over the past few years, is having the unintended consequences of awakening the global community to their existence. The notion of "hidden collections" takes on new meaning while jeopardizing the long-established veil of exclusivity, access, and protected knowledge. The cultural revolution has begun, and there is no going back.

As in the Wizard of Oz when the curtain has been pulled back, countries across the globe have gained deeper awareness and understanding of the central role art plays in human identity. At the core of this transformation is computational culture, which not only allows us to see the unseen, but makes connections between objects and images, while providing a growing body of information about objects and collections. Museums find themselves in a historic moment of re-imaging the museum as one steeped in computational culture, their audiences seeing the arts with a new awareness of their own cultural, historical, and individual identity, many dedicating themselves to changing long-held narratives that have often devalued non-Western cultures.

Emerging in the last few years and accelerating during the two-and-a-half years of COVID-19 [33,34], where people were isolated at home and online alone, museum webpages with many collections and exhibitions have now been digitized and newly revealed to global audiences, offering fresh perspectives on art and museums. Looking ahead, as more artifacts can be viewed online in 3D and interactive modes, new questions arise as physical and virtual artifacts are juxtaposed in physical space and amplified in virtual space.

### 5.1. Cambodia Reclaims Its Looted Heritage

Cambodian sculptures are on view at the Met in Gallery 249 see Figure 9 and have been the subject of a looted art investigation. They became a victim of undervalued non-Western art, telling us so much about Cambodian heritage and how it is reflected in contemporary Cambodian women [35].

These sculptures seem to link with emerging 21st-century aesthetic values, in that the beauty of these works and the women they portray are simply breathtaking. They remind us of the importance of a people to have access to and be able to see and experience the art of their heritage and country in terms of both cultural/national and individual identity, and for developing a love of self and of one's culture.

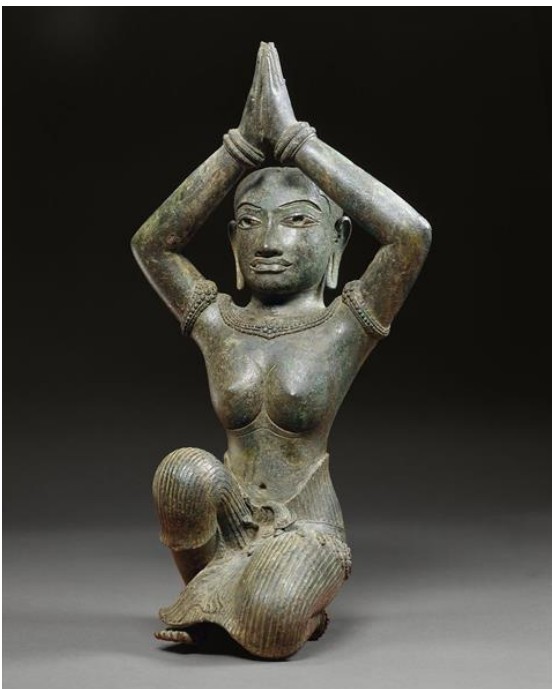 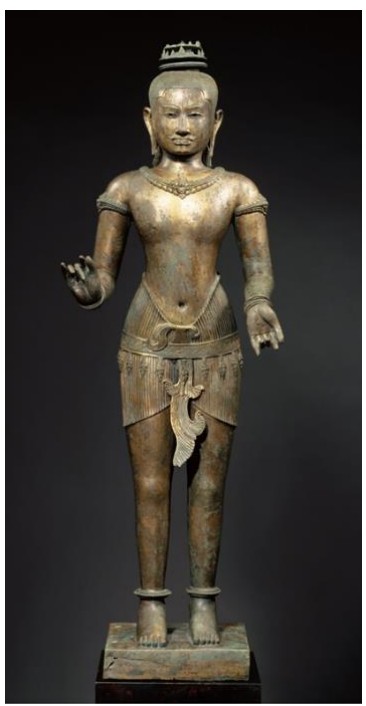

**Figure 9.** Two Cambodian sculptures on view in Gallery 249, Metropolitan Museum of Art, New York. (**left**): Kneeling Female, Bronze inlaid with silver, traces of gold, second half of the 11th century. Perhaps a Khmer queen kneels in adoration, suggesting she was part of an ensemble venerating her deity. The pupils and brows are inlaid, perhaps with black glass. (**right**): Standing Shiva, Cambodia, 11th century, Gilt-copper alloy, silver inlay. Met Museum. https://www.metmuseum.org/art/collection/search/39096 and https://www.metmuseum.org/art/collection/search/39097 (accessed on 30 January 2023).

The ongoing case between the Met and the government of Cambodia symbolizes the broad dimensions of similar cases. This case is representative of others that are pending in the courts. The looting of Cambodian art on a grand scale is a consequence of the Vietnam War [36]; only now is Cambodia dealing with the looting with its main suspect, the Met Museum and the disgraced art dealer Douglas Latchman [37], although some, but not all, material has been returned to Cambodia [38,39]. Now, at a time when non-Western art has achieved a new recognition of its intrinsic value for its spiritualism and living aura, the Cambodian government is seeing the restitution of a significant number of artworks. We see that restitution is largely tied to politics, be it Nazi loot or war, as with Vietnam, or the general devaluing of so-called ethnographic culture that is used to justify the theft of another culture—a type of power equation. To many in Cambodia, the artworks are not just stone artefacts. They embody the spirit of Cambodian culture, its history, and its spirit.

*5.2. Iranian Arts Heritage and Women's Digital Identity*

In our computational world, given that we exist between real and virtual states, there is a preponderance of our hours spent in digital states of being, so that digital identity seems to overshadow physical identity, blurring our sense of time and space, being a click away from tracing our individual heritage and seeing it portrayed in the arts. In this arena, Google Arts and Culture is making important contributions through its collaborative programs with museums in the realm of accessing digital heritage globally.

Digital identity embodied by human digital behavior evolves with an accelerating pace and seems to dominate computational life from social media to Metaverse avatars, intensifying issues from race, ethnicity and gender to the arts and politics [12,14]. Recent mass protests by Iranian women erupted when 22-year-old Mahsa Amini was murdered by the so-called "morality police" for not wearing a headscarf in public. In an unprecedented

turn of events, the Iranian regime backed down, ending the brutal reign of its morality police, but still has not rescinded the 1983 law making the headscarf obligatory. Yet, looking back at Iranian heritage across centuries, we see beautiful, exquisite women with no headscarves; rather we see women bathers in Iranian heritage art. See Figure 10 for an 18th-century example of Iranian women in heritage, now in the collections of the Brooklyn Museum, also found on Wikimedia Commons. Clearly, this was a protest about individual freedom, human rights and identity that instantly reverberated globally over social networks, raising questions about how Iran's heritage present to past might change its future [40]:

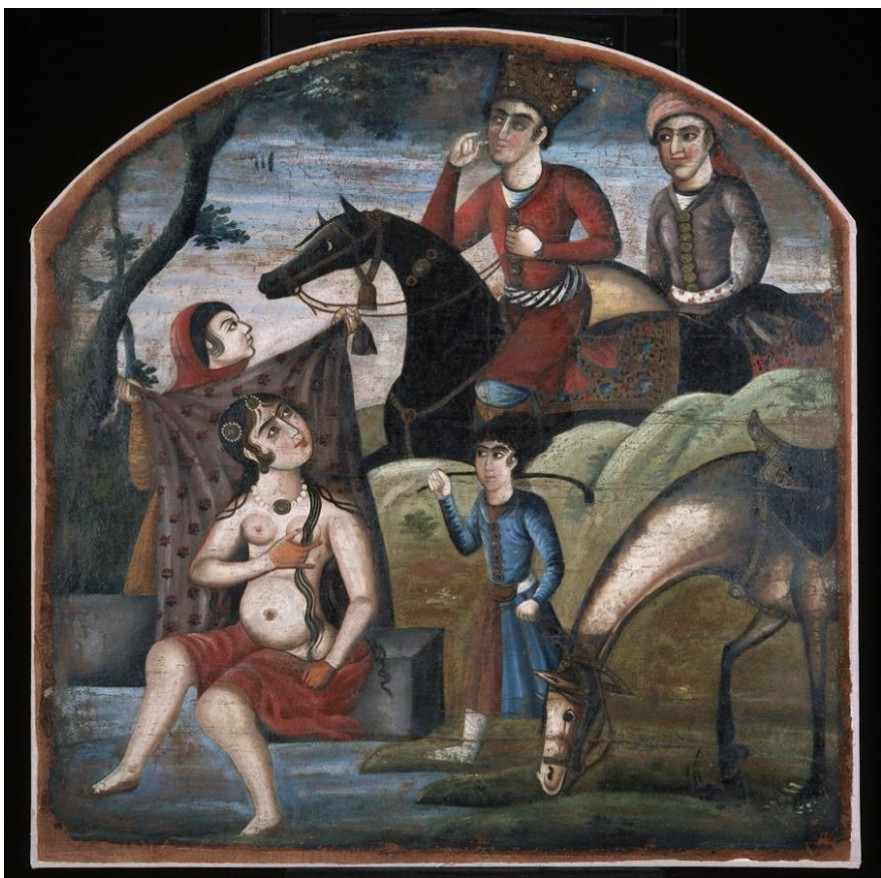

**Figure 10.** Khusraw Discovers Shirin Bathing, from Pictorial Cycle of Eight Poetic Subjects, oil on canvas, 18th century, Iran, collection of the Brooklyn Museum, with a note that, "the tale of Khusraw and Shirin found its greatest expression as one of the five narrative poems of the Khamsa (Quintet), composed by the celebrated Nizami of Ganja (1141–1209), one of Iran's greatest masters of romantic poetry", telling a story of love and passion. Wikimedia Commons. https://commons.wikimedia.org/wiki/File:Khusraw_Discovers_Shirin_Bathing,_From_ Pictorial_Cycle_of_Eight_Poetic_Subjects_-_Google_Art_Project.jpg (accessed on 30 January 2023).

> *"Through social media, mobile apps, blogs and websites, Iranian women are actively participating in public discourse and exercising their civil rights", Mahmoudi said. "Luckily for the growing women's rights movements, the patriarchal and misogynistic government has not yet figured out how to completely censor and control the internet".*

Iranian women, fed up with the morality police's heavy-handed approach, have been posting videos of themselves online cutting locks of their hair in support of Amini. Protesters who have taken to the streets have been chanting "Death to the moral police" and "Women, life, freedom" [41].

Women's protests in Iran are about far more than the mandatory headscarf [42], which is symbolic of the larger battle for freedom of expression and human digital identity, in

essence, the right to be authentic to one's aesthetic and sense of self, which intersects with one's heritage. Abbassi's watercolour, *The Lovers* (see Figure 11) beautifully illustrates that in 17th-century Iran, men and women could embrace as an expression of a loving and sensual relationship without the fear, repression, and death being perpetrated by the government of 21st-century Iran [43].

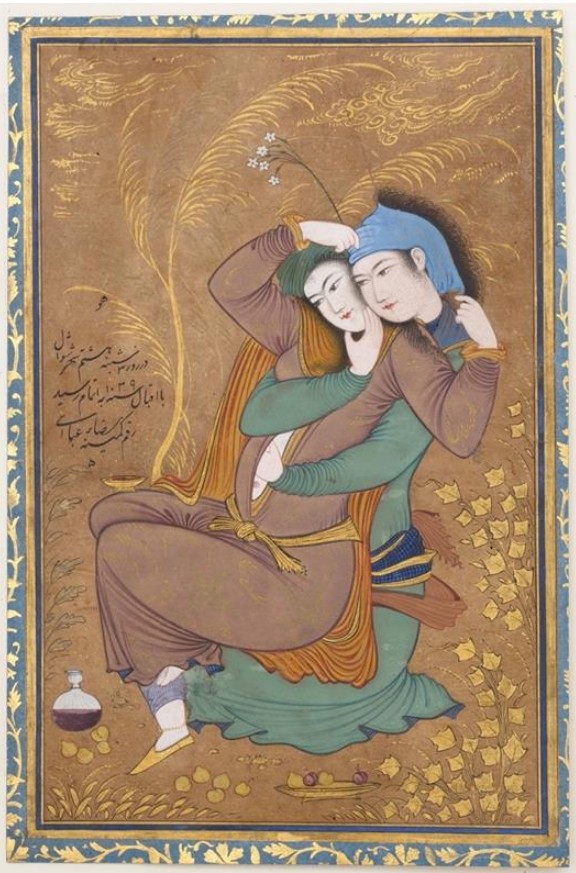

**Figure 11.** The Lovers by Reza Abbassi, Iran, 1630, Opaque watercolour, ink, and gold on paper, Safavid art. Metropolitan Museum of Art. https://www.metmuseum.org/art/collection/search/4510 23 (accessed on 30 January 2023) and https://en.wikipedia.org/wiki/The_Lovers_(Abbasi) (accessed on 30 January 2023).

Iranian works of art (see for example, Figures 10 and 11) are evidential of the status of women in Iran over centuries past; however, in light of the recent women's protests, we see that progress in human rights, although hard-fought, is not guaranteed. After the Iranian Revolution in 1979, could women have predicted that their place in a new society claiming a more advanced culture would come at the price of women's rights and loss of control over their bodies, being subject to the whims of men? On the other hand, gay men were regularly executed, statistics showing that to be the fate of around 5000 men [44].

With the push to digital and online, networks and platforms, especially during COVID-19 [33,34], the Internet became a window into seeing behind the scenes, into places where only the insiders could go, owing to large museum digitization projects. The public can see never-before exhibited museum objects online and watch auctions live, and this has been happening on a global scale. Works of art that were not part of the discussion are suddenly in full view—and more than that—with key advances in AI and technology, such as natural language processing (NLP) and image recognition, with simple searches on Google, users could find buried treasure. Awareness of looted works that have been forgotten began appearing on computer screens worldwide [45].

Museum collections such as those of the Met accumulate and enable the display of heritage from past centuries. The museum still needs to connect with the evolving perspectives of 21st-century society, rather than visiting the past without connections to the present and without re-imagining and reinterpreting art within current contexts to which its audience can relate.

Global searching and global marketing ties to social justice, using new terms and phrases as being "woke" and "looted heritage", has meant that awareness of social movements over the past few years has increased dramatically through, for example, the Internet, social media and museum websites, including digitized collections, online curators, and new narratives expressed around emergent relationships between the arts and sociopolitical movements [45].

### 5.3. Digitizing Heritage and Public Access

The photographic archives of the Johnson Publishing Company (JPC), which include more than three million photographic negatives and slides, 983,000 photographs, 166,000 contact sheets, and 9000 audio and visual recordings, represent the most comprehensive collection documenting black life in the 20th century [46]. Digitizing the Johnson Archive of *Ebony* and *Jet* magazines provides a major resource for black history in America, enabling historians and scholars to bring documentary and photographic evidence conveying fresh perspectives and narratives with detailed historical accounts of the history, arts and culture of black people in America (e.g., see Figure 12). Digitization of these iconic heritage magazines should, we hope, provide open access globally, moving from being "hidden" in the shadows of storage to online and in plain sight. As such, positioned to shine new light on black history at the very moment of intense cultural conflict, it will contribute significantly to advancing racial understanding and dispelling disinformation. If a single picture is worth a thousand words, imagine the value of this archive, that dates from World War II to the Civil Rights Movement of the 1980s and 1990s, and includes some four million images.

Since the 1960s, marking independence for 17 African countries, there were calls for restitution that went unanswered. Currently, we are experiencing restitution at an unprecedented scale. Much credit goes to protestors, activists, and everyday people-pressure in the context of a new sense of global community, social justice, and identity. As public pressure builds, moving from outside to inside the museum, the response is slow and with difficulty, often lacking an understanding of the social and cultural implications as evidenced in the Arts Council England guide to restitution and repatriation. On the face of it, the report by Arts Council England (ACE), titled "*Restitution and Repatriation: A Practical Guide for Museums in England*", might offer some hope on this score [47]. But equally, the many silences in this document and its attempt to focus purely on technical procedure rather than curatorial practice, may signal a new episode in longstanding institutional inertia and resistance.

Museum archives and libraries often play a key role in the documentation of restitution claims, as they hold records with data detailing the acquisition of all objects. Visitor access policies for museum libraries and archives vary greatly, but outside of large institutions, they are very restrictive or simply do not provide public service. Given that documentation of museum objects is critical to art historical research, including ownership, users must turn to digital solutions emerging in our computational world, allowing users to break free from archaic practices of institutional gatekeepers that can act as counter-pressure to new conceptions of open access embedded in the new global cultural order, that challenges who assigns heritage value and ownership. Social justice movements, in the first instance, Black Lives Matter (BLM), inscribed new contexts for asserting cultural identity, spurring new appreciation and recognition of their heritage as the world witnessed the astonishing beauty of African art being restituted.

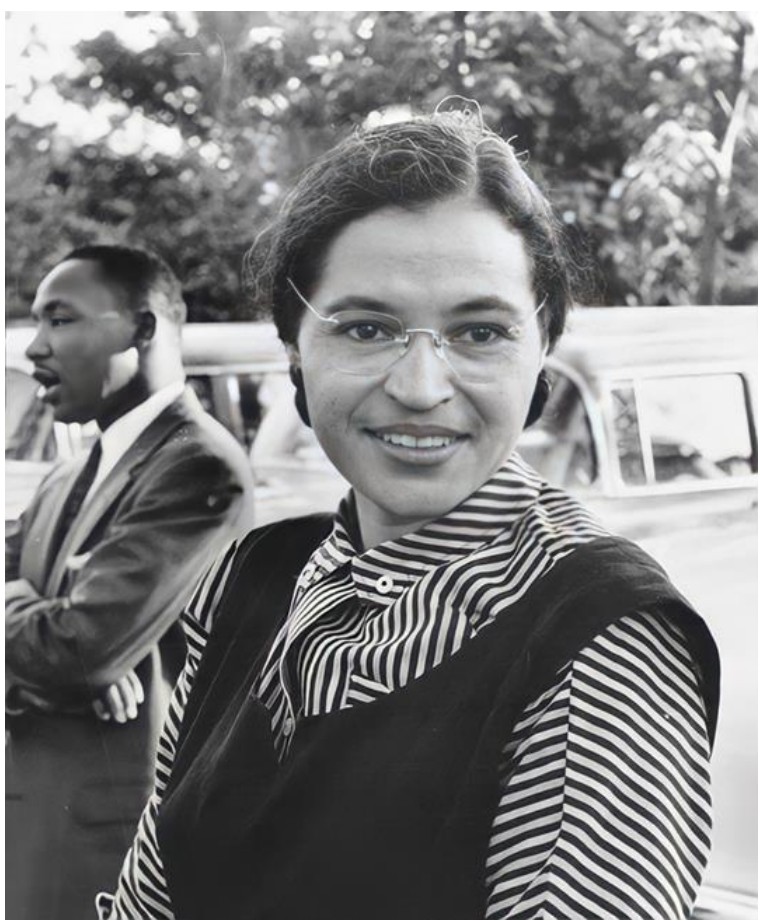

**Figure 12.** Rosa Parks and Dr. Martin Luther King, Ebony Magazine 1955, Mrs. Rosa Parks, Montgomery, Alabama, near the bus boycott. Wikimedia Commons. https://commons.wikimedia.org/wiki/File:Rosaparks_(Remini_enhanced).jpg (accessed on 30 January 2023).

Along these lines, the researchers experienced a pivotal moment memorialized in Paris in December 2018 when they were speakers at the EVA Paris Conference at the Musée Quai du Branly, known for its vast collections of African art. During this very moment, word came concerning the French President, Emanuel Macron, that he had just met with the President of Benin. Having read the report on restitution by Sar and Savoy [48] that Macron commissioned, and which now stands as a landmark text on restitution, Macron pledged to return a number of artworks from Benin from the Musée Quai du Branly collections in Paris. Coincidently, among the very objects that Benin selected to be restituted were ones of exceptional beauty, photographed during this EVA Paris 2019 conference held at the museum [49]. Since 2018, the restitution movement, tied to the growing emphasis on cultural identity, continues to accelerate as part of a global heritage movement [50]. As Macron anticipated, restituted works that are finally being displayed in African museums are attracting large enthusiastic crowds making new connections with their heritage [51].

As more art and archives take their place online, we gain a greater connection to our heritage and digital identity, while concurrently becoming more engaged with the global art community and expanding our appreciation of diverse cultures. Encounters with archives, especially those that have been hidden, can lead to significant discoveries that allow us to re-access history, write new narratives, and sometimes enable researchers to produce popular books [52].

## 6. Conclusions

Touchpoints of human digital identity in our global computational culture are constructed around human interaction in computational contexts, from communicating and sharing thoughts and images with millions of people, experiencing and comparing diverse cultures, to seeing the diversity of world culture, while finding one's digital identity in a world of greater complexity that moves us ever closer to the notion of a diverse global community.

In our computational culture, seeing the Metaverse on the horizon, we find ourselves in a metaphysical place beyond nature and beyond the real, as we live ever more in imagined worlds envisioned through the arts, past and present, while merging heritage and digital states of being; our cultural and digital identity takes its place in mind, body, and soul. Connecting heritage and identity tells us about who we were and have become. It speaks to ways of imagining our future. At a pivotal moment where we begin to write global history, real and virtual identity takes on a more urgent meaning.

During the COVID-19 lockdowns, and now in a post-pandemic mode, museums continue to redesign their websites in ways that serve the visitor experience, education, and knowledge. Importantly, alongside these developments, museums have been advancing their understanding of the critical relationship of cultural heritage to both human and institutional identity, real and virtual. Hopefully, this will dispel museum reticence to embrace the digital in the life of the museum in order to engage fully with their visitors, whose real life is occupied each day with a hefty dose of the digital and perhaps with digital dreams, too.

The accelerating speed of human connections and sharing in the burgeoning virtual landscape makes it more challenging to navigate the natural world and our physical environment where heritage, exposed to climate change, cultural conflict, and diverse digital identities, challenges us to find common ground, as the space between physical and virtual life evaporates into the new reality of being more than human, reflecting the nature of our virtual states of being captured by our pervasive digital identity. We are having our metaphysical moment in a sea of change, as we move from the culture of the melting pot, to one of diversity, equity, and inclusion, where the arts and artists of all stripes and colors are participants, contributing to a new global understanding of heritage and human digital identity.

**Author Contributions:** Conceptualization, T.G. and J.P.B.; methodology, T.G.; formal analysis, T.G. and J.P.B.; investigation, T.G.; resources, writing—original draft preparation, T.G. and J.P.B.; writing—review and editing, J.P.B. and T.G.; project administration, J.P.B.; funding acquisition, T.G. and J.P.B. All authors have read and agreed to the published version of the manuscript.

**Funding:** Jonathan Bowen is funded by the UK Universities Superannuation Scheme (USS).

**Acknowledgments:** Jonathan Bowen received support and additional funding from Museophile Limited. Recent EVA London conference papers on *Electronic Visualisation and the Arts* [12,14] have been inspirational for the background to this paper. Finally, thank you to the reviewers for their help in improving this paper.

**Conflicts of Interest:** The authors declare no conflict of interest. The funders had no role in the design of the study; in the collection, analyses, or interpretation of data; in the writing of the manuscript; or in the decision to publish the results.

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
