# Peer review of "Global Cultural Conflict and Digital Identity: Transforming Museums"

_heritage, doi:10.3390/heritage6020107_

Round 1
Reviewer 1 Report
Overall, the paper proposes an interesting perspective for the scholarly community on the current paradigm shift in museums (and cultural institutions as well) through the lens of global culture and digital identity. In particular, it touches upon urgent topics in the museum sector (and society more broadly), such as ownership and repatriation where the literature is boundless and the discussion implies complexities on different levels (e.g. legal, economic, political, educational as well), which are, more often than we would wish (even if often incidentally), strong limitations to museum’s actions and resolutions, making it a global challenge. Keeping this in mind, the paper discusses these urgent topics mostly effectively, stressing the role of social justice movements and trends as drivers, and making a direct connection between the social subject discussed and the art production/value as an expression of identity, to reinforce their argument.
Like in the example provided, where Florence’s mayor Nardella aspires to make his city creative and innovative, going beyond the static notion of “città-museo”, cultural institutions – and in particular museums – are urged to revise and expand their roles and responsibilities in society and must take advantage of innovation and social trends in computational contexts, actively offering not only physical and digital spaces but the authors suggest a “new reality” where the connection between heritage and identity takes place.
The research method is clearly stated. However, I would suggest adding as a reference other works from peer-reviewed sources beyond their own publications and online sources only.
It is pleasant to read despite the different topics discussed and it needs minor revision only.
In the conclusions, I would expect more discussion on the role of museums in facing the “new reality” above mentioned.
Minor comments:
Line 167. [7,8,9] change full stop with comma.
Line 242. After “hidden” there is a full stop, rather than a comma.
Line 461. Change with "evolves at an accelerating pace".
Line 590. Edit phrase “to seeing the diversity of world culture. while finding one’s digital identity”
Author Response
See attached PDF file.

Reviewer 2 Report
The article does not follow a clear structure. It contains ideas that suggest different themes, each of which could be a different paper. There is a lack of a clear argumentative structure that would help to understand the purposes of the work and therefore to know whether they have finally been achieved.
The description of the methodology is confused and imprecise. An emerging research methodology is mentioned, but no contribution beyond a hermeneutic interpretation of the reference materials is appreciated in the development of the work.
The reference to highly topical events such as the war in Ukraine, as well as other references relating to historical events, do not contribute to clarifying the text, as there is a lack of a deeper reflection that connects them.
There are interesting reflections on the unexpected benefits of the massive digitisation of museum collections, in relation to increasing global conscience about the despoliation of cultural heritage. Perhaps this could be the idea around which to rewrite the paper.
Author Response
See attached PDF file.

Reviewer 3 Report
The paper entitled ‘Global Cultural Conflict and Digital Identity: Transforming Museums’ is very interesting and should be considered for publication. As most of museums are passing through the challenge of improving their digital tools, this article brings an important contribution. Moreover, this paper comes with examples and debates on Ukrainian museums and arts objects which were affected by the recent war and these issues will certainly bring attention from many readers interested in this topic. However, there are a few places in the paper which have to be a little bit reshaped or/and expanded.
First, in the introduction of the paper it should be highlighted what this paper brings new in current international literature on museum studies as well as to highlight what elements of this paper bring added value to what we know in the broader museum studies and in transforming (digital) museum studies in particular.
Second, the literature review in this article is a good one but it has to include several ideas on what we know from previous museum studies and to make the link between classical museums and digital museums. For instance, authors mention very well that ‘Museum archives and libraries often play a key role in the documentation of restitution claims ...’. Here it would be good to make some connections to previous museum studies (see studies such as that of Rhiannon Mason on cultural theory and museum studies, 2006, the book of Carbonell B.M., 2012 entitled Museum Studies-An anthology of contexts’, articles by Light D. el al., 2019, on the educational role of museums published in Southeast European and Black Sea Studies, and another study in journal Societies, 2021 based on assessing the impact of a memorial museum on the young visitors. On the other hand, relationship of transitional justice and memorial museum (see a study of Balcells L. et al., 2022 in the Journal of Politics – doi: 10.1086/714765, see Selimovici M. Johanna’s article on the role of objects in a Balkan museum, 2022) and issues connecting transitional justice to the ‘political work’ of domestic tourism in relation to museums are also important (see an article in Current Issues in Tourism, 2021). Even visitors’ sentiments, such as as empathy on object collections and victims, is also a new trend in museum studies (see doi:10.1080/15387216.2019.1581632).
Third, the methods are nicely presented and innovative, but it must be presented the limitations of method and data used in this paper.
Fourth, the results are nicely presented, but one paragraph of policy recommendations would be great to be added.
Finally, conclusions are short and could be expanded by presenting more the outcomes and the strengths of the paper. Conclusions should also present the international implications of the results of this study or how the outcomes of this paper pushes further what we already know in the existing (digital) museum studies as well as how other studies could further develop the results or the outcomes of this study.
Round 2
Reviewer 2 Report
The paper is a good contribution to the current panorama in which museums are being transformed by the implementation of digital tools. The contribution helps to open up different lines of debate to be taken into account when establishing new museum strategies.
Reviewer 3 Report
Authors have done a very good revision, so I am happy to propose this paper to be accepted for publication.